

# Population genetic structure of Texas horned lizards: implications for reintroduction and captive breeding

Dean A. Williams[1], Nathan D. Rains[2] and Amanda M. Hale[1]

[1] Department of Biology, Texas Christian University, Fort Worth, TX, United States of America
[2] Texas Parks and Wildlife Department, Cleburne, TX, United States of America

## ABSTRACT

The Texas horned lizard (*Phrynosoma cornutum*) inhabits much of the southern Great Plains of North America. Since the 1950s, this species has been extirpated from much of its eastern range and has suffered declines and local extinctions elsewhere, primarily due to habitat loss. Plans are underway to use captive breeding to produce large numbers of Texas horned lizards for reintroduction into areas that were historically occupied by this species and that currently have suitable habitat. We used mitochondrial markers and nuclear microsatellite markers to determine levels of genetic diversity and population structure in 542 Texas horned lizards sampled from across Texas and some neighboring states to help inform these efforts. Texas horned lizards still retain high genetic diversity in many parts of their current range. We found two highly divergent mitochondrial clades (eastern and western) and three major genetic groupings at nuclear microsatellite loci: a west group corresponding to the western mitochondrial clade and north and south groups within the eastern mitochondrial clade. We also found some evidence for human-mediated movement between these genetic clusters that is probably related to the historical importance of this species in the pet trade and as an iconic symbol of the southwestern United States. We do not know, however, if there are fitness costs associated with admixture (especially for the western and eastern clades) or if there are fitness costs to moving these lizards into habitats that are distinctly different from their ancestral areas. If present, either one or both of these fitness costs would decrease the effectiveness of reintroduction efforts. We therefore recommend that reintroduction efforts should maintain current genetic structure by restricting breeding to be between individuals within their respective genetic clusters, and by reintroducing individuals only into those areas that encompass their respective genetic clusters. This cautionary approach is based on the strong divergence between genetic groupings and their correspondence to different ecoregions.

# INTRODUCTION

The loss of suitable native habitat to agriculture and urbanization, and the overexploitation of populations have been the largest drivers of the decline of many species (*Maxwell et al., 2016*). As a result of these and other anthropogenic factors, populations become small and highly fragmented, further endangering species persistence due to stochastic demographic

Corresponding author
Dean A. Williams,
dean.williams@tcu.edu

and genetic effects (*Frankham, Ballou & Briscoe, 2010*). Reintroduction and reinforcement programs try to reduce these effects by returning a species to an area from which it became locally extinct or by increasing the numbers of individuals in small populations (*IUCN, 2013*; *Seddon, 2010*). A sufficient number of individuals with high genetic diversity should be utilized in these efforts to reduce the potential for inbreeding depression and enhance the ability of a population to adapt to changing environmental conditions (*Johnson et al., 2010*; *Carlson, Cunningham & Westley, 2014*; *Jamieson, 2015*). Reintroduction success can be increased by releasing individuals that are matched ecologically and genetically to the introduction region (*Houde, Garner & Neff, 2015*; *Marr et al., 2018*). Additionally, it may also be advisable to prevent mixing of individuals from populations that are ecologically and genetically divergent, to reduce the chances of outbreeding depression (*Frankham et al., 2011*; *Weeks et al., 2011*). Captive breeding programs can potentially be used to raise large numbers of individuals for these reintroduction efforts and are subject to many of the same genetic considerations as reintroductions (*Ebenhard, 1995*; *Williams & Osentoski, 2007*; *Attard et al., 2016*). Understanding the population genetic structure of a species can inform these efforts by identifying appropriate source populations, defining management units, and identifying populations that have high genetic diversity (*Weeks et al., 2011*; *Attard et al., 2016*).

Texas horned lizards (*Phrynosoma cornutum*) belong to a specialized group of lizards (*Phrynosoma*) that are endemic to North America. They have a variety of adaptations for living in dry environments and for specializing on a diet of large venomous ants (e.g., *Pogonomyrmex* spp.; *Sherbrooke, 2003*). Texas horned lizards have an extensive range in North America and cover a number of different ecoregions (*Price, 1990*). Very little genetic work has been conducted on this species, although several studies have found two distinct mitochondrial clades that correspond to a more western clade in New Mexico and Arizona and an eastern clade in Texas (*Guerra, 1998*; *Rosenthal & Forstner, 2014*). The geographic extent of these clades is not clear, however, due to a lack of comprehensive sampling in past studies. Texas horned lizards are generally sedentary and aspects of their life history and anatomy suggest dispersal is relatively limited (*Sherbrooke, 2003*), which could result in strong population structure for this species. On the other hand, anecdotal accounts suggest that these lizards have been moved extensively by the pet trade and individual collectors (*Price, 1990*), leading to more population homogenization than might be expected from natural dispersal.

Within Texas, the species was historically found in nine of ten ecoregions (United States Environmental Protection Agency (EPA) Ecoregions of the US—Level III) and anecdotal accounts suggest it was abundant in many areas (*Price, 1990*; *Donaldson, Price & Morse, 1994*). Declines in Texas began to be noticed between 1950 and 1970, and the species has since disappeared from much of its eastern range with decreases in abundance and local extinctions reported for other areas (*Price, 1990*; *Donaldson, Price & Morse, 1994*; *Henke, 2003*). The major reason for the declines is most likely the loss of suitable habitat due to agriculture and urbanization (*Donaldson, Price & Morse, 1994*). Other possible factors include the introduction of red fire ants (*Solenopsis invicta*) which can prey on the eggs and young of horned lizards, the loss of harvester ants (*Pogonomyrmex* spp.) due to widespread

use of insecticides and competition with fire ants, and over-collecting for the pet and curio trades (*Price, 1990*; *Donaldson, Price & Morse, 1994*). Currently the species is listed as threatened in Texas due to the declines, but globally it is classified as least concern by the IUCN since the species is still common in the more western and southern parts of its range (*Hammerson, 2007*).

Widespread interest by private landowners and the Texas Parks and Wildlife Department has led to a plan in which Texas zoos will captive breed large numbers of individuals to be reintroduced into areas that historically had Texas horned lizards and that currently have suitable habitat. To aid in this effort, we used mitochondrial markers and nuclear microsatellite markers to determine levels of genetic diversity and population structure of Texas horned lizards across Texas and some neighboring states. Species like Texas horned lizards, which occur over large geographic areas, inhabit a range of habitats, and have relatively low dispersal capabilities may have an increased chance of developing regional adaptations (*Lenormand, 2002*). We therefore also ask if genetic subdivisions are related to ecoregions, which could potentially indicate the presence of regionally-adapted units. These results will be used to determine the most appropriate source populations for reintroduction efforts and to provide recommendations for how captive populations should be managed. The neutral genetic patterns described in this study will also help inform future planned studies of adaptive genetic diversity in this species.

## MATERIALS & METHODS

### Sampling and DNA extraction

A number of volunteers collected 542 Texas horned lizard tissue samples across Texas, New Mexico, and Colorado between 2009 and 2017 (Fig. 1). While most of these samples were collected using the cloacal swab method described in *Williams et al. (2012)*, some were from tissues collected from road kill and toe clips collected as part of other population studies. Field activities were approved by Texas Parks and Wildlife Department (SPR-1006-763). Texas Christian University Institutional Animal Care and Use Committee (IACUC) provided approval for this research (protocol 01/08). Samples were mapped onto EPA level II and III ecoregions using ArcGIS Pro and each sample was classified as belonging to a specific ecoregion (*Omernick, 1987*; *Omernick, 1995*; *Omernik & Griffith, 2014*) (https://www.epa.gov/eco-research/ecoregions).

We extracted DNA by incubating swabs and tail or toe clips overnight in 300 μl lysis buffer and 15 μl of Proteinase K (20 mg/ml) at 55 °C. The following day, a half volume of 7.5 M ammonium acetate was added to precipitate proteins. Then 0.7 volume of isopropanol was added to the supernatant and the samples were placed at −20 °C overnight to precipitate the DNA. Finally, DNA was pelleted and washed in 70% ethanol, air dried, and resuspended in 100 μl 10 mM Tris pH 8.5.

### Genotyping and sequencing

We amplified 10 microsatellite loci in three multiplexes using 10 μl polymerase chain reactions (PCR; *Williams et al., 2012*). Four loci (*PcGATA49, PcGATA61, PcGATA60, PcGATA31*) reported in *Williams et al. (2012)* gave evidence of high levels of null alleles

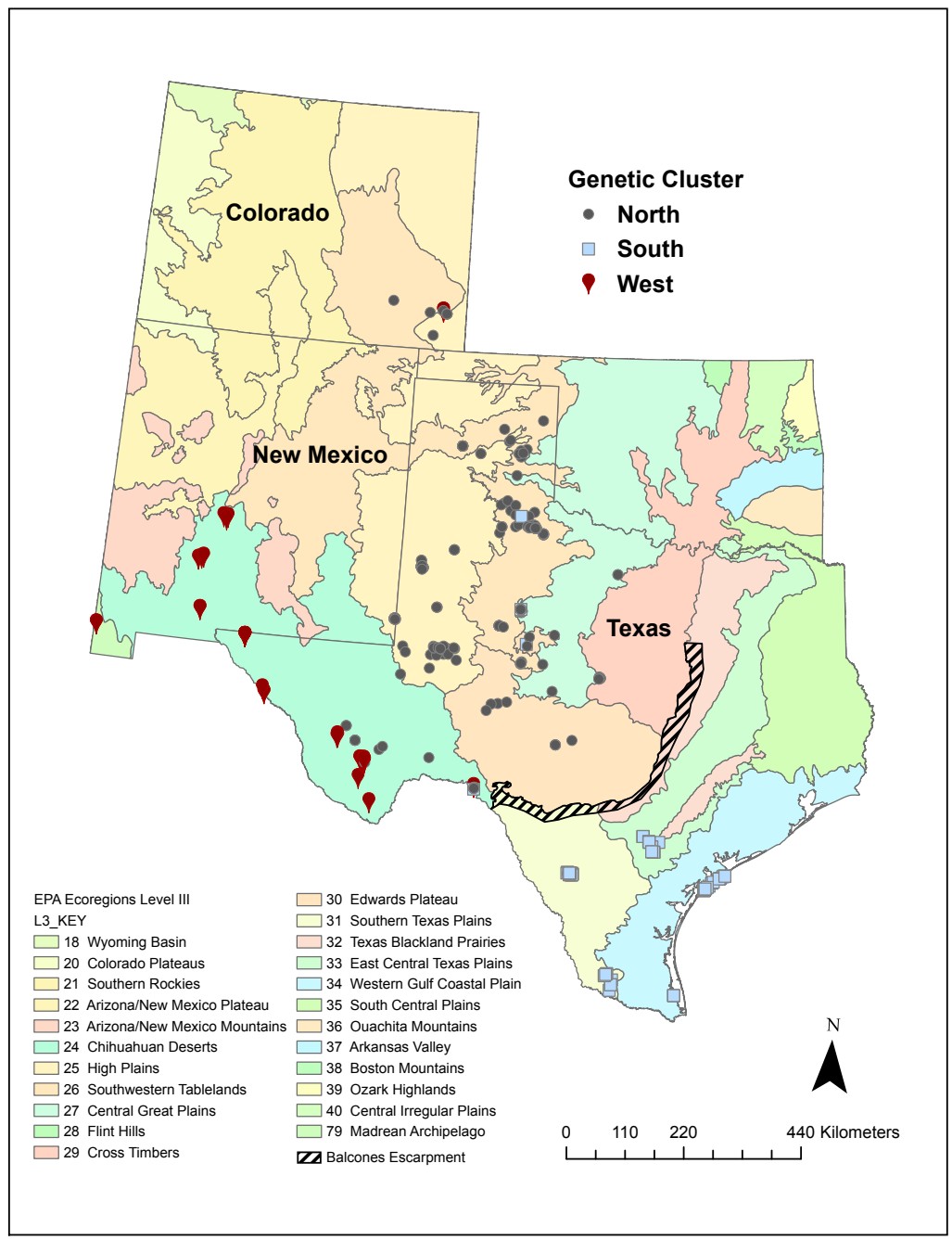

**Figure 1** Sampling locations of 542 Texas horned lizards, *Phrynosoma cornutum*, within EPA level III ecoregions (https://www.epa.gov/eco-research/level-iii-and-iv-ecoregions-continental-united-states). Symbols indicate assignment ($q > 0.49$) to different genetic clusters or populations (west, north, and south) based on multilocus microsatellite genotypes using the program STRUCTURE.

in multiple populations, or problems with large allele dropout (*PcGATA 49*) and so are not included in this study. PCR reactions contained 10–50 ng DNA, 0.2 μM of each primer, 1× Qiagen Multiplex PCR Master Mix with HotStarTaq, Multiplex PCR buffer with 3 mM $MgCl_2$ pH 8.7, and dNTPs. Reactions were cycled in an ABI 2720 thermal cycler (ThermoFisher Scientific, Waltham, MA, USA). The cycling parameters were one cycle at 95 °C for 15 min; followed by 35 cycles of 30 s at 94 °C, 90 s at 60 °C, 90 s at 72 °C; then a final extension at 60 °C for 30 min. Following amplification, all reactions were diluted with 200 μl $dH_2O$. For each sample, 0.5 μl of diluted product was loaded in 10 μl HIDI formamide with 0.1 μl LIZ-600 size standard (ThermoFisher Scientific, Waltham, Massachusetts, USA) and electrophoresed on an ABI 3130XL Genetic Analyzer (ThermoFisher Scientific, Waltham, MA, USA). Genotypes were scored and binned using Genemapper 5.0 (ThermoFisher Scientific, Waltham, Massachusetts, USA). We reamplified a subset of samples (∼10%, $n = 55$ individuals) to estimate the genotyping error rate.

We amplified a 353 bp fragment near the 5′ end of the mitochondrial control region using primers PcCR F—CTTATGATGGCGGGTTGCT and PcCR R—GGCTGTTAAATTTAT CCTCTGGTG for all 542 individuals. We also amplified the mitochondrial NADH dehydrogenase subunit 4 (ND4) and the tRNAs Histidine, Serine, and Leucine region using the primers ND4—ACCTATGACTACCAAAAGCTCATGTAGAAGC and Leu— CATTACTTTTACTTGGATTTGCACCA from *Arèvalo, Davis & Sites (1994)* in 49 individuals from across the sampled range. PCR reactions contained 10–50 ng DNA, 0.25 μM of each primer, 1× Qiagen Multiplex PCR Master Mix with HotStarTaq, Multiplex PCR buffer with 3 mM $MgCl_2$ pH 8.7, and dNTPs. Reactions were cycled in an ABI 2720 thermal cycler. The cycling parameters were one cycle at 95 °C for 15 min; followed by 35 cycles of 30 s at 94 °C, 15s at 55 °C, 30 s at 72 °C; then a final extension at 72 °C for 5 min. Reactions were cleaned enzymatically with *ExoI* and *rSAP* using the manufacturer's protocols (New England Biolabs Ipswich, Massachusetts, USA). Products were sequenced in both directions using PCR primers and internal primer ND4#2-TACGACAAACAGACCTAAAATC from *Arèvalo, Davis & Sites (1994)* for ND4 and electrophoresed on an ABI 3130XL Genetic Analyzer. Sequences were trimmed, edited, and put into contigs using Sequencher 4.8 (Gene Codes Corporation, Ann Arbor, MI USA). Sequences were aligned in MEGA 10 (*Kumar et al., 2018*) using Clustal W (*Thompson, Higgins & Gibson, 1994*). All unique sequences have been deposited in GenBank (accession numbers: MK100594–MK100710).

## Microsatellite genotype analyses

The cryptic nature and low density of Texas horned lizards in most places made it difficult to find large numbers of individuals from single localities. To calculate traditional genetic diversity metrics, however, we grouped 449 of the 542 horned lizards into 16 sampling sites with ≥10 individuals in each (Table 1, Fig. 2). We grouped lizards into these sites because we were interested in estimating the genetic diversity present in protected areas such as Wildlife Management Areas and State Parks where individuals might be captured for future captive breeding purposes. In addition to these nine protected areas, we also chose seven other areas such as counties with ≥10 sampled lizards, to increase our number of sampling sites to test for isolation-by-distance. The coordinates for these sites were the centroids of

**Table 1** Mean ± SE genetic diversity measures at ten microsatellite loci for 449 Texas horned lizards, *Phrynosoma cornutum*, from 16 sampling sites (each with ≥ 10 individuals).

| State | Site | Sampling site name | Cluster | N | $N_A$ | $A_R$ | $H_O$ | $H_E$ | $F_{IS}$ |
|---|---|---|---|---|---|---|---|---|---|
| Texas | 1 | Brewster Co. | W/N | 31 | 13.00 ± 1.50 | 8.73 ± 0.67 | 0.79 ± 0.04 | 0.83 ± 0.03 | 0.05 ± 0.04 |
| | 2 | Hueco Tanks SP | W | 12 | 8.20 ± 0.84 | 7.64 ± 0.77 | 0.79 ± 0.05 | 0.78 ± 0.04 | −0.02 ± 0.03 |
| | 3 | Seminole Canyon SP | N | 17 | 11.10 ± 0.82 | 9.16 ± 0.62 | 0.86 ± 0.03 | 0.84 ± 0.03 | −0.03 ± 0.03 |
| | 4 | Midland Co. | N | 30 | 13.20 ± 1.86 | 8.80 ± 0.75 | 0.83 ± 0.03 | 0.84 ± 0.03 | 0.02 ± 0.01 |
| | 5 | Camp Bowie | N | 11 | 8.00 ± 0.56 | 7.75 ± 0.52 | 0.86 ± 0.04 | 0.81 ± 0.02 | −0.05 ± 0.04 |
| | 6 | Mitchell Co. | N | 14 | 9.40 ± 1.28 | 8.16 ± 0.93 | 0.83 ± 0.03 | 0.83 ± 0.02 | −0.01 ± 0.03 |
| | 7 | Grey Co. | N | 11 | 9.40 ± 0.95 | 9.08 ± 0.86 | 0.86 ± 0.05 | 0.86 ± 0.03 | −0.04 ± 0.05 |
| | 8 | Yoakum Dunes WMA | N | 36 | 12.70 ± 1.62 | 8.56 ± 0.72 | 0.84 ± 0.04 | 0.83 ± 0.04 | −0.01 ± 0.01 |
| | 9 | Matador WMA | N | 55 | 14.50 ± 2.10 | 8.62 ± 0.66 | 0.86 ± 0.02 | 0.85 ± 0.02 | −0.01 ± 0.01 |
| | 10 | RPQRR | N | 79 | 15.50 ± 2.20 | 8.60 ± 0.73 | 0.86 ± 0.02 | 0.84 ± 0.03 | −0.02 ± 0.02 |
| | 11 | CMA | N | 20 | 11.30 ± 1.33 | 8.49 ± 0.77 | 0.82 ± 0.06 | 0.81 ± 0.05 | −0.01 ± 0.04 |
| | 12 | Chaparral WMA | S | 63 | 14.50 ± 1.52 | 8.68 ± 0.56 | 0.83 ± 0.02 | 0.87 ± 0.01 | 0.04 ± 0.02 |
| | 13 | Starr Co. | S | 10 | 8.10 ± 0.69 | 8.10 ± 0.69 | 0.82 ± 0.03 | 0.81 ± 0.02 | −0.02 ± 0.04 |
| | 14 | Matagorda Island WMA | S | 30 | 8.90 ± 1.15 | 6.63 ± 0.58 | 0.70 ± 0.05 | 0.79 ± 0.02 | 0.11 ± 0.05 |
| Colorado | 15 | Colorado | N | 13 | 10.40 ± 0.85 | 9.63 ± 0.73 | 0.87 ± 0.03 | 0.86 ± 0.02 | −0.01 ± 0.03 |
| New Mexico | 16 | E. New Mexico | N | 18 | 11.30 ± 1.21 | 8.90 ± 0.86 | 0.85 ± 0.06 | 0.81 ± 0.05 | −0.04 ± 0.03 |

**Notes.**

Site column is the number corresponding to map locations; cluster column is the population determined by STRUCTURE to which each site belongs: W, west; N, north; S, south; N is the number of individuals sampled at each site; $N_A$ is the number of alleles; $A_R$ is allelic richness standardized for 10 individuals; $H_O$ is observed heterozygosity; $H_E$ is expected heterozygosity; and $F_{IS}$ is the inbreeding coefficient.

For sampling site names, SP, state park; WMA, wildlife management area; CMA, Cross Bar Management Area; Co., county; RRQRR, Rolling Plains Quail Research Ranch.

the sampled lizard locations. For these 16 sites, we used GenAlEx 6.5 (*Peakall & Smouse, 2006*; *Peakall & Smouse, 2012*) to calculate observed ($H_O$) and expected heterozygosity ($H_E$) and $F_{IS}$. We used HP-RARE (*Kalinowski, 2005*) to calculate allelic richness ($A_R$) at each site to standardize comparisons across sample sizes. We tested for heterozygote excess and deficiencies and genotypic linkage disequilibrium using GENEPOP 4.0 (*Rousset, 2008*). We used sequential Bonferroni correction to determine significance for these tests. MICRO-CHECKER 2.2.3 (*Van Oosterhout et al., 2004*) was used to determine the presence of null alleles, large allele dropout, or issues with scoring due to stuttering and to calculate null allele frequencies (*Brookfield, 1996* eq1). As there was some evidence of null alleles, we used the ENA correction method from *Chapuis & Estoup (2007)* implemented in the software FreeNA to calculate global and pairwise $F_{ST}$. We used these corrected values to test for isolation-by-distance using the Mantel test in GenAlEx. Because the magnitude of $F_{ST}$ is influenced by heterozygosity, we also present the standardized measure $F'_{ST}$ developed by *Meirmans & Hedrick (2011)*. We also used a principal component analysis (PCA) in GenAlEx to visualize the pattern of pairwise $F'_{ST}$ between sampling sites and compared these to the STRUCTURE results (see below) which utilized all samples.

We used STRUCTURE 2.3.4 (*Pritchard, Stephens & Donnelly, 2000*) to cluster multilocus genotypes from all 542 samples, including those not found within the 16 sampling locations

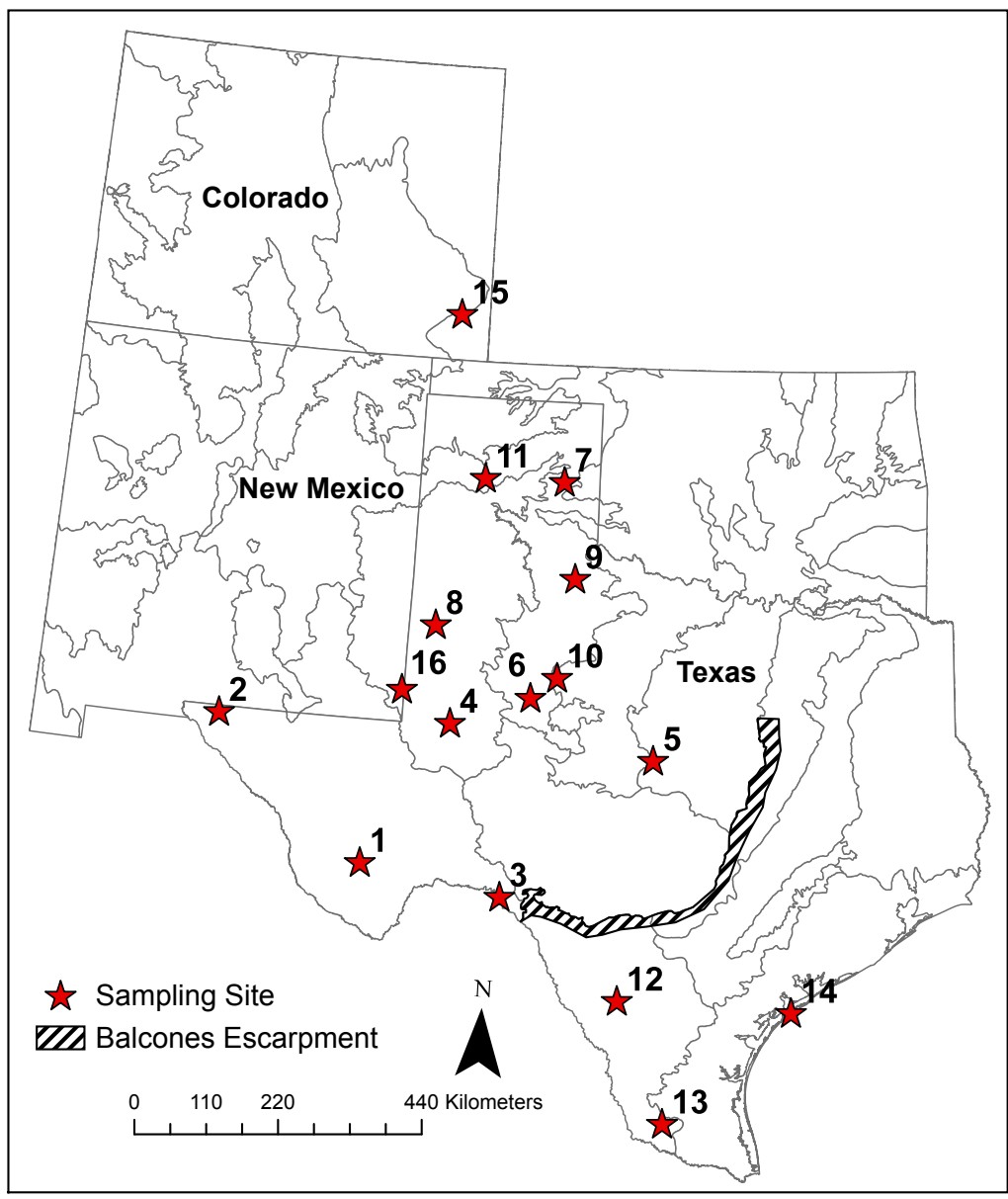

**Figure 2** Locations of 16 sampling sites with ≥10 individual Texas horned lizards, *Phrynosoma cornu-tum*. Numbers are, 1, Brewster Co.; 2, Hueco Tanks SP; 3, Seminole Canyon SP; 4, Midland Co.; 5, Camp Bowie; 6, Mitchell Co.; 7, Grey Co.; 8, Yoakum Dunes WMA; 9, Matador WMA; 10, RPQRR; 11, CMA; 12, Chaparral WMA; 13, Starr Co.; 14, Matagorda Island WMA; 15, Colorado; 16, E. New Mexico.

mentioned above. We assumed admixture and correlated allele frequencies with no location prior and set the burn-in to $10^4$ iterations and ran the MCMC (Monte Carlo Markov Chain) for $10^6$ iterations. STRUCTURE can give misleading results both for the number of populations and individual ancestry if there is uneven sampling across clusters (K; *Puechmaille, 2016*; *Wang, 2017*). We used the recommendations of *Wang (2017)* and set the prior for admixture to allow $\alpha$ to vary between clusters and we decreased the initial

$\alpha$ from 1.0 to 0.2. We ran ten independent runs for $K = 1$–10. The most likely $K$ was identified using the method of *Evanno, Regnaut & Goudet (2005)* and by determining the $K$ with the highest LnP(D) before values started to plateau (*Pritchard, Stephens & Donnelly, 2000*). We then used CLUMPP 1.1.1 (*Jakobsson & Rosenberg, 2007*) to average across the ten runs for the most likely K. We considered individuals to be admixed between clusters when their ancestry ($q$) was $\geq 0.10$ in each of two or more clusters. We chose this value since similar cut-off values have been used in a number of other studies (*Barilani et al., 2005*; *Vähä & Primmer, 2006*; *Sanz et al., 2009*; *Bohling, Adams & Waits, 2013*; *Johnson et al., 2015*). The null alleles we found in this study are not expected to impact the accuracy of these assignments given the very small difference in $F_{ST}$ values after correction (see 'Results') and because simulation studies have found only a slight reduction in the power of assignment tests in the presence of null alleles (*Carlsson, 2008*; *Marsh et al., 2008*).

## Mitochondrial sequence analyses

We used GenAlEx to calculate the frequency of haplotypes and haplotype diversity ($h$) of the control region for the 16 sampling sites mentioned above. We used CONTRIB 1.02 (*Petit, Mousadik & Pons, 1998*) to calculate haplotype richness ($H_R$) at each site. We compared mtDNA population structure for these 16 sites to the nuclear microsatellite structure by calculating $\phi_{PT}$ for both data sets and conducting AMOVAs in GenAlEx.

We constructed single gene trees for both ND4 and the control region due to the difference in sampling for each marker. The model for nucleotide substitution was that with the highest value of the Bayesian information criterion (BIC) in MEGA. The resulting models were HKY + G for ND4 and HKY + G + I for the control region. These models were then used to create maximum likelihood trees in MEGA using the nearest-neighbor-interchange and a strong branch swap filter. Bootstrap values were calculated using 1,000 replicates. *Phrynosoma asio* (JN809342.1) was used to root the ND4 gene tree and *Phrynosoma blainvillii* (NC_036492.1) was used to root the control region tree. Bayesian analyses using MrBayes 3.2 (*Huelsenbeck & Ronquist, 2001*; *Ronquist et al., 2012*) were also conducted for the ND4 region. The analysis began with two parallel runs with random starting trees and were run for $2 \times 10^7$ generations and sampled every $10^3$ generations. Burn-in was the first 25% of generations and a 50% majority rule consensus tree was calculated.

We evaluated whether well-supported Texas horned lizard clades (bootstrap $\geq 80$) identified in the gene trees had experienced past population expansion or were stable, using all samples sequenced for the ND4 region and a subset of samples for the control region. We used DnaSP 6.11.01 (*Rozas et al., 2017*) to calculate the frequency of haplotypes, haplotype diversity ($h$), nucleotide diversity ($\pi$), and $\theta$ (Watterson's estimator). We also conducted neutrality tests in DnaSP, including Tajima's D (*Tajima, 1989*) and Fu's $F_S$ (*Fu, 1997*) to compare the number of rare and common mutations to the null hypothesis of a stable population, in contrast to an expectation of an excess of low frequency mutations following rapid population growth. When D and $F_S$ were significantly different from neutral expectations for the ND4 data, we used the mismatch distribution to estimate the time since population growth, tau ($\tau$) as $\tau = 2 \mu kt$, where $t$ is the time in generations,

μ is the mutation rate per site per generation, and k is the sequence length (*Rogers, 1995*). We calculated tau in Arlequin 3.5.1.3 (*Excoffier & Lischer, 2010*). We assumed a generation time of two years (*Ballinger, 1974*), and used an online calculator to calculate time since population growth (*Schenekar & Weiss, 2011*). We used a substitution rate of 0.00805 (substitutions/site/million years) which was estimated for ND4 in geckos (*Macey et al., 1999*) and used by *Blair & Bryson (2017)* in their study of species delimitation in *Phrynosoma*.

## RESULTS

### Microsatellite data

Most loci across the 16 sampling sites were in Hardy-Weinberg equilibrium and all loci were in genotypic linkage equilibrium. There was one locus/site comparison (out of 160) that had a significant heterozygote deficit after sequential Bonferroni correction. In addition to this one comparison, MICRO-CHECKER also identified five other loci/site comparisons that had null alleles (Table S1). There were four loci that gave evidence of null alleles at one site each and another locus that had evidence for null alleles at two sites (Table S1). There were a total of 19 individual/locus non-amplifications in the entire dataset, spread across six of the loci. We found eight allele errors across 1,140 alleles in the 55 individuals we re-genotyped for an error rate of 0.007. These errors occurred across five loci (per locus error rates for these five loci ranged from 0.009 to 0.018). A total of seven samples (12%) had one allele error each.

Genetic diversity was generally high (mean ± SE: $H_E = 0.83 \pm 0.006$, $A_R = 8.47 \pm 0.18$, $n = 16$ sampling sites) and similar across sites, with the exception of the population on Matagorda Island which had lower heterozygosity and allelic richness than mainland sites (Table 1). We corrected for null alleles when calculating $F_{ST}$; however, this made virtually no difference in the observed patterns ($F_{ST}$ values generally only decreased by 0.001). Population differentiation was modest but significant with a global $F_{ST}$ of 0.044 (95% CI [0.033–0.061]) and pairwise $F_{ST}$ ranging from 0.003 to 0.115 (Tables S2 and S3). Standardized values were much higher with a global $F'_{ST}$ of 0.302 and pairwise values ranging from 0.011 to 0.665. The first two axes of the PCA of pairwise $F'_{ST}$ explained 52.4% of the variance and revealed a tight cluster of 11 sites (RPQRR, Midland Co., CMA, Yoakum Dunes WMA, Matador WMA, Seminole Canyon SP, Grey Co., Mitchell Co., Camp Bowie, Colorado, E. New Mexico) and two other more separated sets of sites (Chaparral WMA, Starr Co., Matagorda Island WMA) and (Brewster Co., Hueco Tanks SP; Fig. 3). Most pairwise $F_{ST}$ comparisons were significantly different from zero except for 23 comparisons which were all between sites within the cluster of 11 sites in the PCA (Table S2). There was a significant pattern of isolation-by-distance for all 16 sampling sites ($R^2 = 0.33$, $P = 0.003$) and for the 11 sampling sites that formed the tight cluster in the PCA ($R^2 = 0.24$, $P = 0.02$).

The STRUCTURE analysis using all samples revealed three major genetic clusters or populations, both using the *Evanno, Regnaut & Goudet (2005)* method and the LnP(D) method of *Pritchard, Stephens & Donnelly* (*2000*; Fig. S1). At $K = 3$, samples were partitioned into west, north, and south populations (Fig. 4). At increasing levels of K (4–10),

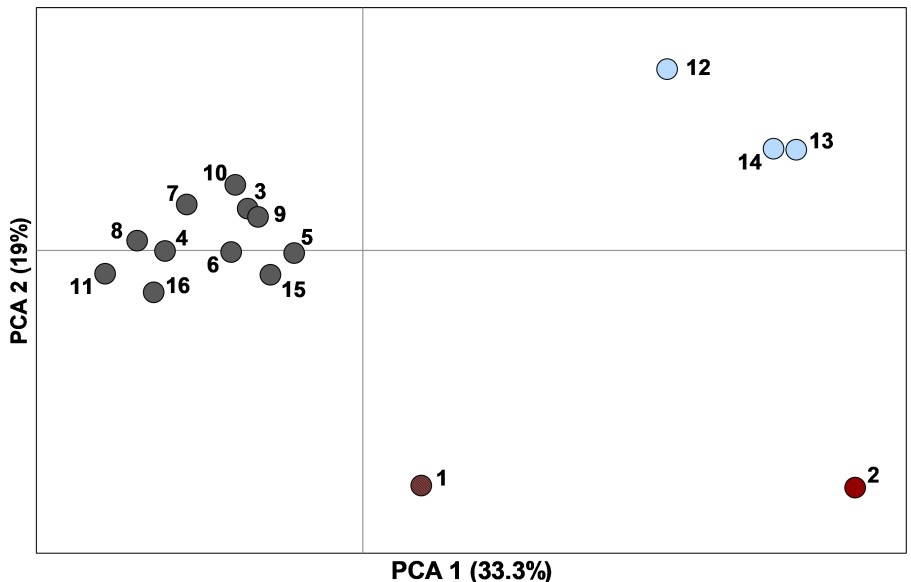

**Figure 3** Principal component analysis of pairwise F′<sub>ST</sub> values determined with ten nuclear microsatellites, between 16 sampling sites (*n* = 449 individuals) with ≥10 Texas horned lizards, *Phrynosoma cornutum,* each. Colors correspond to populations determined by STRUCTURE: dark red, west; light blue, south; dark grey, north, dark grey and red hatch marks on #1 indicates a mix of north and west. Numbers are sampling sites: 1, Brewster Co.; 2, Hueco Tanks SP; 3, Seminole Canyon SP; 4, Midland Co.; 5, Camp Bowie; 6, Mitchell Co.; 7, Grey Co.; 8, Yoakum Dunes WMA; 9, Matador WMA; 10, RPQRR; 11, CMA; 12, Chaparral WMA; 13, Starr Co.; 14, Matagorda Island WMA; 15, Colorado; 16, E. New Mexico.

new clusters were simply added as admixture in the large north population. Sub-clustering the south population produced a split between Matagorda Island WMA and the mainland (Fig. S2). Sub-clustering the west population revealed two clusters, one composed of west individuals and another population from Brewster Co. around the Elephant Mountain WMA (Site #1 in Fig. 2, Fig. S2). Most of the individuals (83%; 25 of 30) assigned to the west sub-population had western clade haplotypes (see mitochondrial section below), whereas most of the individuals (95%; 21 of 22) assigned to the Elephant Mountain WMA sub-population had eastern clade mitochondrial haplotypes. Sub-clustering the north population did not reveal more populations. The north, south, and west populations found by the STRUCTURE analyses were concordant with the pattern of pairwise population F′<sub>ST</sub> seen in the PCA. The 11 sites that clustered close together all belonged to the north population, whereas the Chaparral WMA, Starr Co. and Matagorda Island WMA belonged to the south population. Hueco Tanks SP was assigned to the west population and Brewster Co., which fell between the northern population and Hueco Tanks SP in the PCA, had individuals assigned to north and west populations and individuals that were admixed between the populations.

The west population is found within the Chihuahua Deserts and Madrean Archipelago ecoregions and the south population is found south of the Balcones Escarpment in the Southern Texas Plains, East-Central Texas Plains, and Western Gulf Coastal Plain

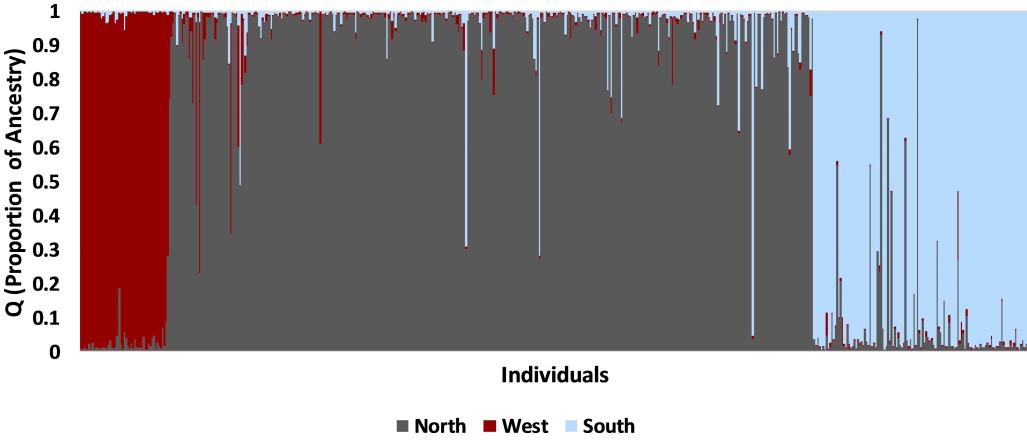

**Figure 4** Bayesian clustering of nuclear multilocus microsatellite genotypes from 542 Texas horned lizards, *Phrynosoma cornutum*, using STRUCTURE for *K* = 3. Each vertical line indicates the proportion of ancestry (*q*) for an individual lizard with the colors representing the cluster or population identified in STRUCTURE. Individual lizards are organized by geographic sampling location, starting with the most western locations on the left and then moving to more northern locations and then southern locations.

**Table 2** Average proportion of ancestry (*q*) by ecoregion, as determined in STRUCTURE using 10 microsatellite markers for 542 Texas horned lizards, *Phrynosoma cornutum*. Shading has been added to illustrate the ecoregions with the highest ancestry in each genetic cluster (West, North, South).

| Ecoregion level II | Ecoregion level III | West | North | South |
|---|---|---|---|---|
| South-Central Semi-Arid Prairies | High Plains | 0.01 | 0.98 | 0.01 |
| | Southwestern Tablelands | 0.02 | 0.95 | 0.03 |
| | Central Great Plains | 0.01 | 0.94 | 0.05 |
| | Edwards Plateau | 0.02 | 0.89 | 0.10 |
| | Cross Timbers | 0.01 | 0.98 | 0.01 |
| Southeastern USA Plains | East-Central Texas Plains | 0.01 | 0.06 | 0.94 |
| Tamaulipas-Texas Semiarid Plain | Southern Texas Plains | 0.01 | 0.11 | 0.88 |
| Texas-Louisiana Coastal Plain | Western Gulf Coastal Plain | 0.00 | 0.02 | 0.98 |
| Western Sierra Madre Piedmont | Madrean Archipelago | 0.98 | 0.02 | 0.00 |
| Warm Deserts | Chihuahua Deserts | 0.62 | 0.36 | 0.02 |

ecoregions (Table 2, Fig. 1). The north population is found north and north-west of the Balcones Escarpment within a number of level III ecoregions including the High Plains, Central Great Plains, Southwestern Tablelands, Edwards Plateau, and Cross Timbers which collectively belong to the level II South-Central Semi-Arid Prairies ecoregion (Table 2, Fig. 1). Some individuals with genotypes from the north population (*n* = 28 individuals) also occurred in the Chihuahua Deserts ecoregion in Texas. There were two individuals that had high (*q* > 0.90) ancestry assignment in the north population but were found in the South Texas Plains, and one individual with high ancestry assignment in the south population that was found in the Central Great Plains ecoregion.
Most individuals (90%; 486 of 542) were strongly assigned to a single population. The remaining individuals (10%; 56 of 542) had evidence of admixture. Admixture between the south and north populations ($n = 41$ of 488 individuals) was more common than between west and north populations ($n = 13$ of 421 individuals; Fisher exact test, $P = 0.001$). The remaining two individuals were admixed between all three populations; one was found in southern Texas whereas the other was found in northern Texas. Admixed individuals with north and west ancestry were found in Brewster Co. where the north and west populations meet ($n = 4$ individuals) and in widely separated areas including Colorado ($n = 4$), Seminole Canyon SP ($n = 3$), Matador WMA ($n = 1$), Yoakum Dunes WMA ($n = 1$), CMA ($n = 1$), and Starr Co. ($n = 1$). Admixed individuals with north and south ancestry were found south of the Balcones Escarpment ($n = 14$) in the south population and far north of the Escarpment ($n = 27$) in the north population.

**Mitochondrial data**

Haplotype diversity at the control region was high with a total of 86 haplotypes found across all 542 individuals. Haplotype diversity ($h$) and richness was more variable between the 16 sites than microsatellite diversity (Tables 1 and 3). For instance, Midland, Yoakum Dunes WMA, and RPQRR had relatively low haplotype diversity and richness compared to their microsatellite diversities. Despite the variability in mitochondrial diversity there was a positive correlation between haplotype richness and allelic richness ($r_S = 0.57$, $P = 0.02$). Population subdivision was considerably higher for the mtDNA control region than for the microsatellite loci with 32% of the variance between sites for the control region ($\phi_{PT} = 0.32$, AMOVA, $P = 0.001$) and 8% of the variance between sites for the microsatellite loci ($\phi_{PT} = 0.08$, AMOVA, $P = 0.001$).

The control region gene tree revealed two clades between haplotypes found in western areas (New Mexico and far western Texas) and more eastern areas (Fig. 5). The western clade was well supported in this analysis (bs = 80). For the control region, there were a total of ten unique haplotypes in the western clade and 76 haplotypes in the eastern clade. There was an average of 3.91% divergence between the clades (range: 2.01–5.31%), 1.43% divergence within the eastern clade (range: 0.28–2.92%), and 0.08% divergence within the western clade (range: 0.28–1.72%). *P. cornutum* differed from *P. blainvillii* by 12.40% (range: 11.31–13.74%). The western haplotypes mainly belonged to individuals with high microsatellite ancestry in the west population (mean ± SE q: 0.97 ± 0.003, $n = 28$), with the exception of two individuals that had high north ancestry ($q = 0.99$) and were found in Brewster Co. Within Brewster Co., there were also 21 individuals with high west ancestry ($q > 0.90$) that had eastern haplotypes, indicating that this region is an area of admixture between the western and eastern mitochondrial clades. We found no evidence for separate mitochondrial clades corresponding to the north and south populations detected by STRUCTURE. Individuals with the eastern haplotypes had microsatellite ancestry within the north, south, and west populations. Control region haplotypes were, however, largely unique to the geographic regions encompassed by the three main nuclear microsatellite populations (west, north, and south) as indicated by the colored symbols in Fig. 5. Only seven of the 86 haplotypes were shared between regions. The south region had the highest

**Table 3** Mitochondrial diversity at the control region for 449 Texas horned lizards, *Phrynosoma cornutum*, from 16 sampling sites (each with 10 individuals).

| State | Site | Sampling site | Cluster | N | H | $H_R$ | $h$ |
|---|---|---|---|---|---|---|---|
| Texas | 1 | Brewster Co. | W/N | 31 | 11 | 5.26 | 0.87 |
| | 2 | Hueco Tanks SP | W | 12 | 2 | 1.00 | 0.49 |
| | 3 | Seminole Canyon SP | N | 17 | 9 | 5.58 | 0.89 |
| | 4 | Midland Co. | N | 30 | 4 | 1.23 | 0.25 |
| | 5 | Camp Bowie | N | 11 | 3 | 1.91 | 0.56 |
| | 6 | Mitchell Co. | N | 14 | 5 | 3.35 | 0.73 |
| | 7 | Grey Co. | N | 11 | 7 | 5.46 | 0.82 |
| | 8 | Yoakum Dunes WMA | N | 36 | 3 | 0.67 | 0.11 |
| | 9 | Matador WMA | N | 55 | 12 | 4.12 | 0.75 |
| | 10 | RPQRR | N | 79 | 7 | 2.47 | 0.47 |
| | 11 | CMA | N | 20 | 4 | 2.45 | 0.66 |
| | 12 | Chaparral WMA | S | 63 | 18 | 6.76 | 0.89 |
| | 13 | Starr Co. | S | 10 | 10 | 8.00 | 0.98 |
| | 14 | Matagorda Island WMA | S | 30 | 1 | 0.00 | 0.00 |
| Colorado | 15 | Colorado | N | 13 | 3 | 2.00 | 0.73 |
| New Mexico | 16 | East New Mexico | N | 18 | 10 | 6.47 | 0.85 |

**Notes.**

Site column is the number corresponding to map locations; cluster column is the population determined by STRUCTURE to which each site belongs: W, west; N, north; S, south; N is number of individuals sampled at each site; H is number of haplotypes; $H_R$ is haplotype richness standardized for 10 individuals; and $h$ is haplotype diversity.

For sampling site names, SP, state park; WMA, wildlife management area; CMA, Cross Bar Management Area; Co., county; RRQRR, Rolling Plains Quail Research Ranch.

haplotype diversity ($h = 0.90$), followed by the west ($h = 0.88$), and then the north region ($h = 0.72$). To compare the three regions, we used the standardized estimate of $\phi$PT using the method of Hedrick to ensure that $\phi$PT $= 1.0$ when populations have non-overlapping sets of haplotypes (*Meirmans & Hedrick, 2011*). $\phi$PT was 0.97 ($P = 0.001$) among the three regions, reflecting the low sharing of haplotypes between them.

The ND4 gene tree also revealed two well-supported clades between haplotypes found in western areas (New Mexico and far western Texas) and more eastern areas (Fig. 6). Both the maximum likelihood and Bayesian analyses recovered the same tree topology. There were a total of ten unique haplotypes in the western clade and 21 unique haplotypes in the eastern clade. There was an average of 7.07% divergence between the clades (range: 6.50–7.80%), 0.07% divergence within the eastern clade (range: 0.10–1.70%), and 0.04% divergence within the western clade (range: 0.10–0.60%). *P. cornutum* differed from *P. asio* by 16.40% (range: 15.80–16.80%). Similar to the control region, the western clade comprised individuals with high microsatellite ancestry in the west population. Similar to the control region, we did not find separate clades for the ND4 gene that corresponded to the north and south populations detected by STRUCTURE.

Using all samples sequenced for the ND4 region and a subset of samples sequenced for the control region (same samples as the ND4 region plus all western clade samples), haplotype diversity and sequence diversity were higher for the eastern clade than the western clade (Table 4). This result is consistent with the larger geographic range of the

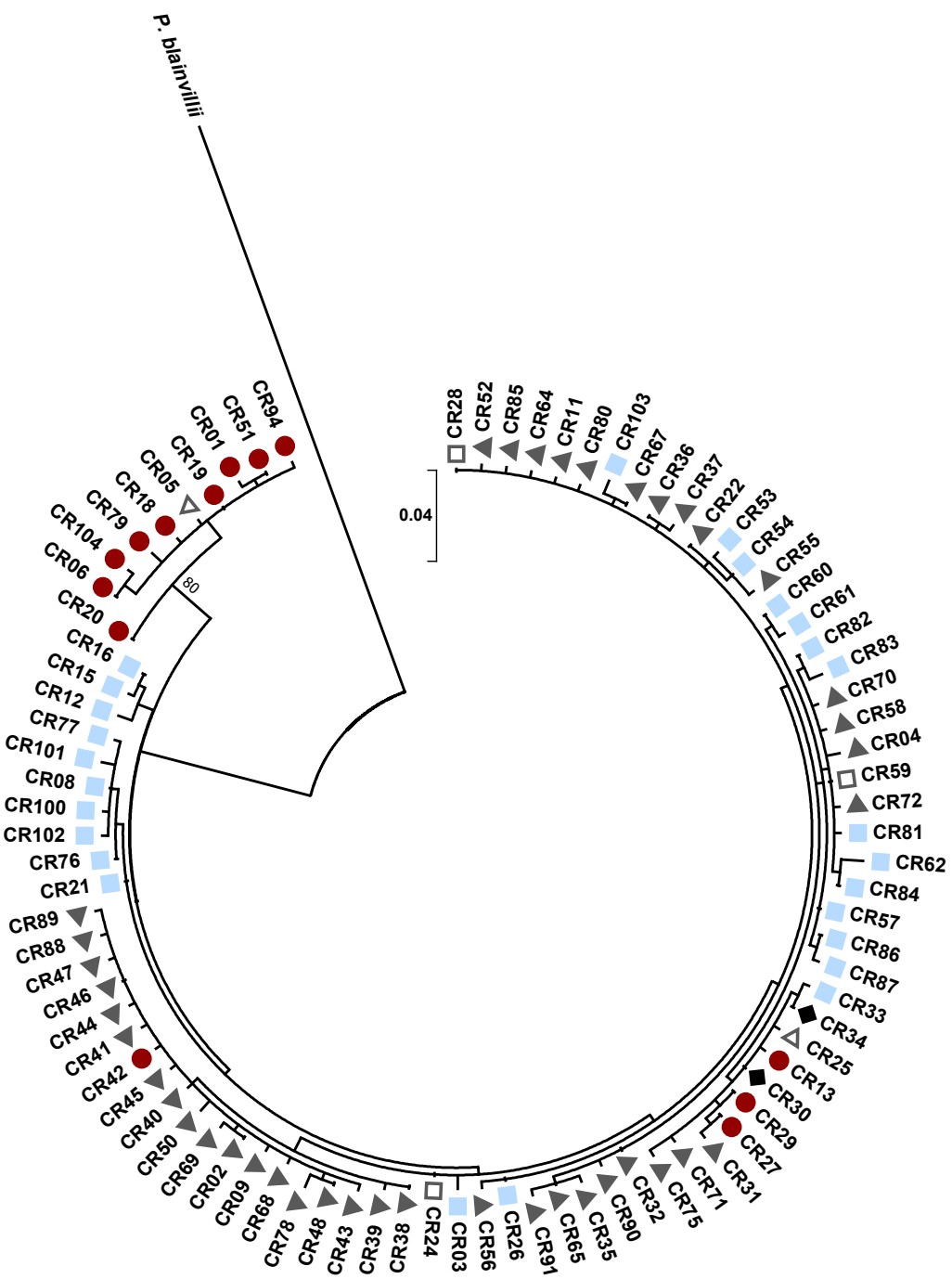

**Figure 5 Maximum likelihood tree for Texas horned lizard, *Phrynosoma cornutum,* mitochondrial control region (353 bp) haplotypes (*n* = 542 individuals) rooted with *Phrynosoma blainvillii.*** Numbers at nodes are bootstrap values (bs). The tree has one well supported clade (bs = 80) corresponding to western localities, with 31 individuals in the western clade and 189 individuals in the eastern clade. Colored shapes indicate the nuclear microsatellite population(s) in which a particular mitochondrial haplotype was found (see text). Red circles indicate the west population, light blue squares the south population, dark gray triangles the north population, open squares both the south and north populations, open triangles both the west and north populations, and black diamonds the west, north, and south populations.

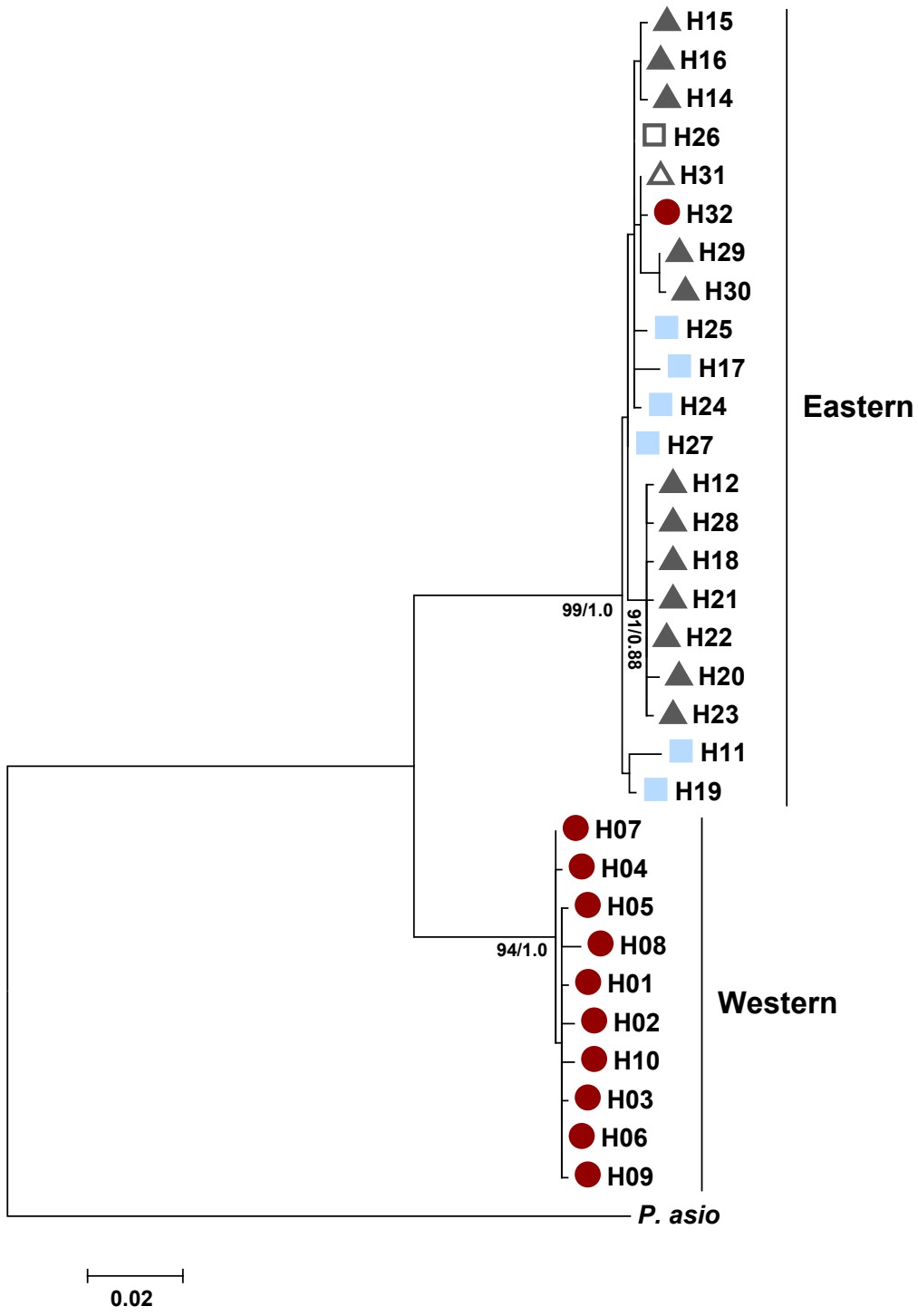

**Figure 6  Maximum likelihood tree for the Texas horned lizard, *Phrynosoma cornutum*, (*n* = 49 individuals) based on 778 bp of the mitochondrial NADH dehydrogenase subunit 4 (ND4) and the tRNAs Histidine, Serine, and Leucine, rooted with *Phrynosoma asio*.** Numbers at nodes are maximum likelihood bootstrap values/Bayesian posterior probabilities. Colored shapes indicate in which nuclear microsatellite population(s) a particular haplotype was found (see text). Red circles indicate the west population, light blue squares the south population, dark gray triangles the north population, open squares both the south and north populations, and open triangles both the west and north populations.

**Table 4** Mitochondrial diversity and demographic estimators for Texas horned lizard, *Phrynosoma cornutum*, western and eastern clades based on 778 bp of the mitochondrial ND4 gene and 353 bp of the mitochondrial control region.

| Clade | N | H | $h$ | $\pi$ | Tajima's D | Fu's Fs | $\theta$ | $\tau$ |
|---|---|---|---|---|---|---|---|---|
| ND4 Western | 16 | 10 | 0.87 | 0.0024 | −2.044[**] | −6.192[***] | 3.918 | 1.895 |
| ND4 Eastern | 33 | 21 | 0.96 | 0.0059 | −1.631[*] | −11.003[***] | 8.131 | 5.852 |
| CR Western | 31 | 11 | 0.75 | 0.0051 | −1.455 | −4.693[**] | 3.254 | – |
| CR Eastern | 33 | 18 | 0.92 | 0.0104 | 0.206 | −8.541[***] | 3.450 | – |

**Notes.**

N, number of individuals; H, number of haplotypes; $h$, haplotype diversity; $\pi$, nucleotide diversity; $\theta$, Wattersons estimator; $\tau$, expansion parameter.

[*]$P < 0.05$.
[**]$P < 0.01$.
[***]$P < 0.001$.

eastern clade and the higher effective population size of the eastern clade as indicated by larger theta values (Table 4). We found evidence for past population expansions for both clades; Fu's $F_S$ was significantly negative ($P < 0.05$) relative to neutral expectations for both western and eastern clades of ND4 and the control region. Tajima's D was significantly negative ($P < 0.05$) only for the ND4 data (Table 4). Using the data from the ND4 region, the time of expansion was estimated to be 151,288 years ago for the western clade and 467,196 years ago for the eastern clade.

## DISCUSSION

With the exception of the Matagorda Island population, the Texas horned lizard sampling localities included in this study harbor high levels of genetic variation and would therefore provide suitable source individuals for captive breeding and reintroduction efforts. Across the surveyed area, Texas horned lizards exhibit two major groupings at mtDNA loci (western and eastern) and three major genetic groupings at nuclear loci (west, north, and south). These major genetic patterns should be used to inform both captive breeding and reintroduction strategies for this species.

The major genetic groupings were found in ecoregions which differ from each other in patterns of precipitation, temperature, and vegetation (*Griffith et al., 2007*). The western mitochondrial clade and population is found in the Chihuahua desert ecoregion, whereas the eastern mitochondrial clade can be subdivided into two nuclear DNA populations that correspond to a north cluster found in the South-Central Semi-Arid Prairies ecoregion (EPA level II) and a south cluster found in the South Texas Plains, East-Central Texas Plains, and Western Gulf Coastal Plain. The Chihuahua desert ecoregion (west clade) is an area of high biodiversity and endemism. The vegetation is predominantly semi-desert grassland and shrubland, with a single rainy season during the summer (late June-October). Precipitation is lower than in the other ecoregions. The South-Central Semi-Arid Prairies ecoregion (northern cluster) encompasses five level III ecoregions and is composed of tall and short grass prairies. Precipitation varies across this region from very low in the High Plains ecoregion to higher precipitation in the Central Plains and Cross Timbers ecoregions. These ecoregions can experience strong seasonality in temperatures, with colder and longer

winters, especially in the more northern areas. The South Texas Plains, East-Central Texas Plains, and Western Gulf Coastal Plains (south cluster) are lower in elevation and have a subtropical climate with mild winters and a pattern of bimodal precipitation occurring in the spring and fall. Thorny brush and coastal grasslands are the predominant vegetation types. Differences in vegetation, precipitation, and temperature between the ecoregions suggest that studies of behavioral differences (e.g., diet and activity budgets), morphological differences important for predator avoidance (e.g., coloration, length of horns) or water regulation (e.g., overall body size, scale size), and physiology (e.g., thermal tolerance, water balance) should be conducted between and within these clusters to determine the presence and scale of potential adaptations in this species.

The eastern and western mitochondrial split in Texas horned lizards was consistent with some previous studies (*Guerra, 1998*; *Rosenthal & Forstner, 2014*), although the eastern extent of the western clade was not clear in these studies due to a lack of sampling in south-western Texas. The presence of a late Pliocene pluvial lake, Lake Cabeza de Vaca, which covered up to 23,000–26,000 km$^2$ in southern New Mexico, far western Texas, and Mexico has been hypothesized as the barrier that originally separated these two clades (*Guerra, 1998*; *Rosenthal & Forstner, 2014*). A number of amphibian, reptile, and mammal species have their eastern or western range limitations in the region of the lake as well as evidence for species or subspecies differentiation on either side of the lake region (*Axtell, 1977*). Phylogenetic studies have also found high divergence on either side of the lake region for several reptile species (*Rosenthal & Forstner, 2014*). The lake system started to drain in the mid-Pleistocene (∼750,000 years ago; *Strain, 1966*; *Reeves, 1965*; *Reeves, 1969*). The estimated time of expansion that we found for the western clade of the Texas horned lizard occurred later in the mid-Pleistocene (151,288 years ago) which may have then spread into far western Texas where it occurs today. The eastern clade's expansion was also estimated to have begun in the middle Pleistocene, but the timing was a few hundred thousand years earlier than what was estimated for the western clade. The higher haplotype diversity found in the south region may indicate that it was the source for expansion into more northern areas.

We recommend that the western and eastern populations, as identified by the mitochondrial analyses presented here, should be considered separate evolutionary significant units (ESUs; *Moritz, 1994*; *Crandall et al., 2000*). There is reciprocal monophyly and high divergence between the mitochondrial western and eastern clades that is also broadly concordant with the nuclear microsatellite data. The western group is also confined to the Chihuahua desert ecoregion which represents a much different habitat than what is inhabited by most individuals from the more eastern group. The mtDNA genetic distance between these clades (7%) is much higher than within-clade distances and may indicate either subspecies or species differences (*Guerra, 1998*; *Rosenthal & Forstner, 2014*). Nonetheless, the presence of admixed individuals in this dataset provide some evidence for hybridization between horned lizards from the western and eastern groupings. This result, in conjunction with evidence that the two clades successfully hybridize in captive breeding programs in zoos (D Williams, pers. obs., 2018) might argue against separate full species designations for eastern and western clades of the Texas horned lizard. We do not know,

however, whether the offspring from these pairings experience lower fitness in the wild. If there is a reduction in fitness, this could support full species delimitation for these clades and warrants further investigation. Interestingly, it appears that most gene flow between the eastern and western clades has been male-mediated. Of the 38 individuals that had evidence of admixture between the two clades, 95% (36 of 38) had a nuclear signature of the west cluster and an eastern mitochondrial haplotype. There were only two individuals with north nuclear ancestry and western clade mitochondrial haplotypes located in Brewster Co. Whether this pattern is related to hybrid incompatibilities (i.e., Haldane's rule; *Haldane, 1922*) is unknown and requires further study.

The microsatellite data detect the western mitochondrial clade and also split individuals in the eastern mitochondrial clade into north and south populations. The most likely dispersal barrier between the south and north populations is the Balcones Escarpment (Fig. 1). The Balcones Escarpment extends through central Texas, from the southwest to the northeast, and is a series of cliffs, hills, and plateaus reaching up to about 300 m in height (*Abbott & Woodruff Jr, 1986*). This geologic feature separates the more xeric habitat of the Edwards Plateau from the more mesic lowlands of the South Texas Plains and East-Central Texas Plains. Individuals to the north of this feature have high ancestry in the north population, whereas individuals south of the escarpment have predominantly south ancestry. Furthermore, only four of 74 control region haplotypes found in these two populations are shared across this potential barrier. A number of reptiles and amphibians in Texas have their eastern and western range boundaries along this escarpment (*Smith & Buechner, 1947*). For example, of the 23 species of lizards that have ranges reaching the Balcones Escarpment, only two species are found on both sides of the barrier: *P. cornutum*, the focus of this study, and *Sceloporus olivaceus*. Phylogeographic studies on snakes, lizards, salamanders, and rodents have also found evidence for genetic breaks across this potential barrier (*Chippindale et al., 2000*; *Castoe, Spencer & Parkinson, 2007*; *Neiswenter & Riddle, 2010*; *Andersen & Light, 2012*; *Moseley et al., 2015*; *Cox et al., 2018*). Our genetic data suggest that this barrier may also limit dispersal for Texas horned lizards, but perhaps not to the degree that has been detected in some of these other studies.

The two main microsatellite populations (north and south) within the eastern clade could also be considered separate management units (MUs; *Moritz, 1994*), based on their differentiation at microsatellite loci and limited overlap in mtDNA haplotypes indicating dispersal constraints between these clusters. These two populations are also found in different habitats.

Dispersal distances (i.e., from nest to area of first breeding) are currently unknown for Texas horned lizards, but radio-telemetry studies, anatomy, and life history characteristics indicate that the species is generally sedentary and probably has limited long-distance dispersal capabilities. Typical home ranges are 0.4–7.0 ha for adult horned lizards and daily movement distances are 0–247 m, although distances up to ∼800 m over several days of travel have been detected in a few cases (*Fair & Henke, 1999*; *Burrow et al., 2001*; *Stark, Fox & Leslie, 2005*; *Endriss, 2006*; *Wall, 2014*; *Mitchell, 2017*). Horned lizards have a flat, tank-like body form that makes them fairly slow and easy to capture once detected. As a result of this body form, Texas horned lizards rely mainly on crypsis and remaining

immobile for long periods of time to avoid detection by predators (*Sherbrooke, 2003*). The presence of an isolation-by-distance pattern in the microsatellite data across all sampling sites, as well as for sites within the northern population, is also consistent with limited dispersal (this study). Population structure as measured by $\phi_{PT}$ was higher for the mitochondrial locus than for nuclear microsatellites. Stronger population structure at mtDNA compared to nuclear loci may also be an indication that populations have recently become fragmented and isolated. Mitochondrial loci might be expected to reveal the effects of drift first since they have a lower effective population size than nuclear loci. The relatively low mtDNA haplotype diversity seen at some sampling sites (Table 3) may be indicative of isolation of those sites.

Texas horned lizards are unique among lizards in Texas because of their nostalgic and symbolic status in historical accounts and folklore (*Welch, 1993*; *Manaster, 1997*; *Sherbrooke, 2003*). Anecdotal accounts suggest that these lizards have been moved extensively by people, and as a result, this species has become locally established outside its native range in coastal areas in the south-eastern United States (*Price, 1990*). For example, a single pet store in the 1950s reported exporting as many as 50,000 Texas horned lizards per year to various areas in the US (*Dropkin, 2015*). Texas horned lizards were also given out for free at some gas stations in Texas with the purchase of a full tank of gas, and they were traded extensively among boys at Boy Scout Jamborees (*Welch, 1993*; *Manaster, 1997*; *Dropkin, 2015*). Even today, the authors of this study have on multiple occasions been told by well-intentioned horned lizard enthusiasts that they have intentionally translocated horned lizards many kilometers to put them on their property or on the property of a friend or relative, or that they simply let them loose, far from where they had been found, after trying to keep the lizards as pets.

The widely separated and isolated occurrences of some admixed individuals is consistent with human-mediated movement from anecdotal accounts. If the admixture we observed was due simply to natural dispersal, we would have expected to find admixture primarily in regions where the microsatellite clusters come into contact. For instance, the west and north populations come into contact in the Chihuahua desert ecoregion in Brewster County, TX where we found evidence for admixture from the microsatellite and mitochondrial loci. We were not able to determine if there was evidence for admixture along the Balcones Escarpment where the north and south populations might be expected to come into contact, as this species has been extirpated from most of the eastern and southern border of the escarpment and only a few horned lizards were sampled directly north of the escarpment in the Edwards Plateau. The presence of some west ancestry in widely separated individuals within the geographic range of the north and south microsatellite populations, as well as the presence of south ancestry in northern Texas (e.g., at the Matador WMA) or north ancestry in southern Texas (e.g., at the Chaparral WMA) seems to be at odds with natural dispersal patterns.

## CONCLUSIONS

Reintroductions of the Texas horned lizard are only planned for areas that historically had horned lizards, currently have suitable habitat, and are within the geographic ranges encompassed by the north and south populations. We recommend that breeding facilities at Texas zoos keep individuals from the three microsatellite genetic clusters separate and return their offspring to regions that correspond to their microsatellite genetic cluster. This strategy is based on the indication that these lizards may be regionally-adapted. Assuming reintroductions are successful, there may also be nearby populations that will eventually come into contact with these introduced individuals. Avoiding the mixing of differentiated clusters would therefore be advisable to reduce the chances of outbreeding depression. Nonetheless, recent reviews have suggested that fears of outbreeding depression may have been over-emphasized in some conservation programs (*Frankham et al., 2011*; *Frankham et al., 2017*; *Ralls et al., 2018*). This may be especially true for instances of genetic rescue in which small populations are experiencing inbreeding depression and need to be augmented to increase genetic diversity (*Frankham, 2015*; *Frankham, 2016*; *Ralls et al., 2018*). For Texas horned lizards, there is the luxury of taking a more cautionary approach, since this species is still abundant with high genetic variation in many areas within each of our defined genetic clusters. The presence of the west-east mitochondrial divide and the fact that microsatellite clusters cover distinct ecoregions further supports that this more cautionary approach is warranted for this species (*Frankham et al., 2011*).

On the other hand, one might argue that genetic contamination has already occurred between these populations given our evidence of admixture in unexpected areas. Selection is expected to eventually remove the deleterious effects of outbreeding depression (*Edmands et al., 2005*; *Erickson & Fenster, 2006*), and it is therefore possible that the admixed individuals we found were a product of that selection. We still do not know, however, if there are fitness costs associated with admixture (especially for the western and eastern clades) or if there are fitness costs to moving these lizards into habitats that are distinctly different from their ancestral areas. If present, either one or both of these fitness costs would decrease the effectiveness of reintroduction efforts. In the future, we recommend that studies be conducted to determine if there is evidence for regional adaptations that correspond to the genetic clusters uncovered in this study. Of course, there may be even more fine-scaled, locally adapted units due to the presence of multiple ecoregions within these clusters. In addition to the behavioral and morphological studies suggested earlier, NGS (next generation sequencing) methods could also be used to identify loci which may be under differential selection to better delineate adaptive conservation units in this species (e.g., *Funk et al., 2012*).

## ACKNOWLEDGEMENTS

We thank the large number of volunteers from Texas Parks and Wildlife, Texas Master Naturalists, Horned Lizard Conservation Society, Fort Worth Zoo, Dallas Zoo, San Antonio Zoo, and others that helped us obtain samples. Lee Ann Linam was especially instrumental in organizing the collection effort and helping collect samples. Jeff Bonner, Jason Brewer,

Ruston Hartdegen, Ashley Inslee, Kris Karsten, Richard Kazmaier, Bradley Lawrence, Kathy and John Lupardus, Chad Montgomery, Maureen Morris, Wanda Olszewski, RL Orth, Richard Reams, Chip Ruthven, Don Sias, and Sara Weaver provided large numbers of samples from specific localities for this study. Cory Leach, Emmanuela Mujica, and Megan Raetz helped develop genetic markers and process samples in the laboratory.

### Funding

This work was supported by a grant from the Texas Parks and Wildlife Department's Horned Lizard License Plate fund, grants from the Andrews Institute of Mathematics & Science Education at TCU, and grants from the TCU Research and Creative Activities Fund. The funders had no role in study design, data collection and analysis, decision to publish, or preparation of the manuscript.

### Grant Disclosures

The following grant information was disclosed by the authors:
Texas Parks and Wildlife Department's Horned Lizard License Plate.
Andrews Institute of Mathematics & Science Education at TCU.
TCU Research and Creative Activities Fund.

### Competing Interests

Nathan D. Rains is an employee of the Texas Parks and Wildlife Department.

### Author Contributions

- Dean A. Williams conceived and designed the experiments, performed the experiments, analyzed the data, contributed reagents/materials/analysis tools, prepared figures and/or tables, authored or reviewed drafts of the paper, approved the final draft.
- Nathan D. Rains conceived and designed the experiments, performed the experiments, contributed reagents/materials/analysis tools, authored or reviewed drafts of the paper, approved the final draft.
- Amanda M. Hale conceived and designed the experiments, performed the experiments, contributed reagents/materials/analysis tools, prepared figures and/or tables, authored or reviewed drafts of the paper, approved the final draft.

### Animal Ethics

The following information was supplied relating to ethical approvals (i.e., approving body and any reference numbers):

Texas Christian University Institutional Animal Care and Use Committee (IACUC) provided full approval for this research (protocol 01/08).

### Field Study Permissions

The following information was supplied relating to field study approvals (i.e., approving body and any reference numbers):

Field activities were approved by Texas Parks and Wildlife Department (SPR-1006-763).

## DNA Deposition

The following information was supplied regarding the deposition of DNA sequences:

The mitochondrial sequences are available at GenBank: MK100594–MK100710.

## Data Availability

The raw nuclear microsatellite genotypes, mitochondrial control region haplotypes, GPS coordinates, 16 sampling sites, and proportion of ancestry as determined by STRUCTURE for all individuals, the alignment of unique haplotypes for the mitochondrial ND4 region and the alignment of unique haplotypes for the mitochondrial control region are all available in the Supplemental Files.

## Supplemental Information

Supplemental information for this article can be found online at http://dx.doi.org/10.7717/peerj.7746#supplemental-information.

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
