# Peer review of "Population genetic structure of Texas horned lizards: implications for reintroduction and captive breeding"

_PeerJ, doi:10.7717/peerj.7746_

## Round 0.1 · original submission · Major Revisions

This manuscript describes population structure of the Texas horned lizard using nuclear and mitochondrial markers to characterise 542 individuals from widespread sampling sites. While I do think it will be an important contribution to the literature regarding this species, there are a number of issues that need to be addressed before the manuscript can be accepted for publication. In particular, the figures need substantial edits and some suggested additional analyses may improve this manuscript.

Both reviewers have made useful suggestions that should be addressed. In particular, there are three major issues that require attention. First, the partitioning of samples into 16 sampling sites requires better explanation. Were these samples taken from a particular geographic radius? If so, are the GPS coords of these 16 areas given in Supp Mat the centroids of these sites? If there is not a systematic explanation for these groupings, I agree with the reviewer that the genetic groupings identified by Structure (N=3, or N=5 if sub-structure is considered) should be used instead to estimate summary statistics for populations.

Second, please pay particular attention to the reviewer comments regarding associations with environment. This area of the paper could be greatly improved and it may be worth considering the addition of a specific analysis to understand how environment shapes genetics in this species (e.g. RDA).

Third, the figures need to be amended and both reviewers have made excellent suggestions regarding this point. In general, the information in figures and tables could be better organised and redundancies removed to make these easier to interpret. It would be fantastic to see a combination figure that shows all the results of all three marker types. For example, the dots on the map could include concentric circles showing the CR and ND3 haplotypes, supp data 1 and supp data 2 could be combined into a single file, etc. Also, since there are several data sets presented, please ensure that all figures explicitly state to which data set they refer.

Please pay attention to inconsistencies in terminology (e.g. clade vs cluster).

Sex biased dispersal is mentioned several times but it wasn't clear if sex information was collected. If so, sex biased dispersal could be explicitly tested. If not, please use care in interpreting the data presented. Differences between markers can also due to ploidy differences, for example.

The PCA plot might be more informative if all individuals are included (and colour coded by site), rather than using population averages.

It may be clearer if you use two different naming conventions for the haplotypes from different markers (C1 and N1, instead H1 for both).

Finally, the manuscript left me wondering if there has been so much human movement and the evidence of this persists in the current population, and if there are natural hybrid zones as described, and successful crosses in captivity, maybe mixing these groups isn’t really an issue? This possibility is approached in the end of the manuscript but that message does not carry through to the abstract - It may be worth tempering/qualifying the language there with respect to management advice?

Reviewer 1 ·

Basic reporting

Satisfactory.

Experimental design

Satisfactory.

Validity of the findings

Satisfactory.

Additional comments

This manuscript meets all of the review criteria required for publication at PeerJ. The basic reporting, experimental design, and validity of findings are all satisfactory.

In general, the paper presents novel results for an interesting study system that is of special conservation concern. There are also practical implications from the study in terms of management of natural populations and implementation of captive breeding projects.

The data are sufficient for answer the basic questions of the study. I only have a few concerns for the authors to consider.

Population groupings - The choice of the 16 population groups seems arbitrary and subjective. What is the justification for these groupings, since they guide all of the descriptive statistics and population comparisons? The groupings have labels that are completely separate from the main document figures. There is no way to identify where these 16 places are in relation to one another without searching through supplement materials. The 16 groups in the PCA plot (Figure 2) have no geographic context without a map. If 16 arbitrary groups are used, then they should be plotted on the map in the main article. However, perhaps the pop. gen summary statistical comparisons should be conducted for the populations detected from the microsatellite data (indicating K=3), instead of determined subjectively to be K=16.

MtDNA data - The mtDNA data alignments are not sufficiently described. For example, the program MEGA is used to manipulate sequences, but this program does not conduct multiple sequence alignment. It uses other algorithms (programs) to perform alignments. This is an issue because the ND4 protein coding gene has gaps that are not in multiples of 3 (corresponding to amino acid indels), indicating an alignment problem or perhaps an erroneous sequence.
It is not clear if the mtDNA stats considered the outgroup, or if only the ingroup was used.
There is no real reason why the two mtDNA genes shouldn't be linked, since they share the same genealogy. Perhaps this was a logistical choice because of the different levels of sampling?
Why wasn't an outgroup used for the control region sequence? A complete mitochondrial genome for a coast horned lizard is available on GenBank. The phylogeny for control region seems relatively useless without branch lengths or an outgroup. This phylogeny figure should be redone to match the style of the ND4 phylogeny. The ND4 tree provides adequate information. The circle tree for control region does not.
I recommend adding a Bayesian skyline analysis to estimate population trajectories, a more modern approach that is based on a coalescent population model that is relevant for these types of data.

Structure analysis - what is the justification for designating population membership values of q>0.1 to signify admixture?
The colors for the structure plot do not match the colors for the groups on the maps. There needs to be some coordination between the colors across the various figures.

·

Basic reporting

Overall a well-written and easy to read manuscript. Though generally clear, there were only a couple of points (maybe in Results) where a sentence was confusing. The structure and flow of the article was simple to follow and seemed well set out. However, whilst the appearance of the figures was professional and high-quality, I tended to find a few figures, and associated captions, could be adjusted to improve clarity. Whilst the context and aims of the work are clear, it felt that a little more detail could be included in the Introduction outlining any previous genetic work and understanding of the focal species, so as to further elucidate how this work was to build upon that knowledge and fill a gap in understanding; something that was only touched on very briefly in the Discussion. Furthermore, although the study on the genetic structuring within the focal species seemed to be reasonably sound and thorough enough to address the genetic aims of the work, more could have been done to address the question of any relationship between genetic divisions and ecoregions, which was raised as the secondary aim of the work. Whilst this would not necessarily require additional analyses, further exploration into this element would be desirable.

Experimental design

The authors have conducted a solid study with an impressive number of study animals and comprehensive sampling of the species’ range. Sufficient and clear information was detailed in the methods, and multiple comparable analyses were conducted on different genetic datasets, which is great practice. Despite my initial concerns the genetic sampling used was modest, analyses appear sound and the authors were able to produce relevant results with which they make informed recommendations as to future management strategies, as per the aims of the study. As mentioned in the section above, a small amount of additional background in the Introduction section clearly identifying the knowledge gaps the genetic work in this study would fill compared to prior work would further strengthen this manuscript.

Validity of the findings

The genetic findings of the study are fairly well presented and put in context. Observed patterns are discussed concisely and relevant possible explanations are provided. The conclusions and associated management recommendations made by the authors are clear and reasonable. I would have liked to see the authors further explore the question of potential adaptation of different population units to specific ecoregions than what is currently presented, particularly given it is stated as an aim of the study.

Additional comments

Major missing details:
As raised in the above sections, I had two main areas that stuck out as warranting further detail.
The first, to really ensure the research is placed in the context of the broader research area, a brief mention in the Introduction of the current standing genetic knowledge of the Texas horned lizard would go a long way. There was one sentence (Ln 340-343) in the discussion, which could be raised in the Intro. for further context, e.g. 'there was known to be an east-west divide but sampling was not geographically comprehensive', or 'we only had an idea of mtDNA patterns, nuDNA would be useful to gain better understanding of the genetic diversity of the species'. Even the potential this represents one panmictic population due to the known history of human translocation vs. the general theory that these are low-dispersal critters, could be good to bring up in the Intro., as it then presents two quite contrasting patterns that could be predicted within the study, that can then be compared to the findings.

The second issue was that aim 2 (relation of genetic divisions to ecoregions) was not quite explored to its potential and I was left wanting more. This is a particularly interesting area of investigation and could be done greater justice, even in terms of fleshing out the detail within the text of how these ecoregions differ. There were no methods/analyses laid out to directly test for an association between genetic groups and ecoregions, which are not absolutely necessary, however, testing for correlations of environmental/habitat variables or significant differences in such variables between regions occupied by different genetic clusters is one example of how this could be done. Alternatively, and regardless, I would have liked to see the authors delve into the differences they believe are present between the broad regions occupied by these different clusters. In particular:
- Results section: - Ln 252-262 – This section seems to be the only part of results where ecoregions are discussed in regards to genetic clusters, but there is still no clear indication there is any preference for particular habitat types from the information provided, simply that different clusters occupy multiple different ecoregions, which could simply be geographic.
- Discussion section: - Ln 363 – “…which represents a much different habitat than…” – Could this be elaborated as to how the ecoregions are different, to better justify that the differences may indeed be relevant to these lizards?
- Ln 385-386 – Possibly the clearest point offered that the habitat types may be strongly different.
- Ln 403-405 – Again, what makes these two broad “regions” so different as habitats, rather than simply divided by a barrier?
- Ln 453-455 & Ln 464-465 – I agree that regional adaptation could be an important factor, but more justification is required to convince me how these broad regions differ in consistent ways.

Defining/referring to structure:
It can fast become tricky to define what clusters or clades are being discussed at any stage when patterns within genetic datasets are not 1:1 concordant. Suggestion is to be consistent in terminology and as clear as possible whenever units are discussed, e.g. be explicit when discussing mtDNA clades vs. nuDNA clades. For example:
- Abstract, Ln 32 – perhaps ‘mitochondrial’ genetic groupings, but could see the argument for without.
- Discussion, Ln 294 – “There was no evidence for separate mitochondrial clades…”
Similarly, consider bringing up the regional divisions (North, South, West) as early as Ln 225-229 within the Results, so the reader can follow that there is a consistent pattern of clustering throughout the analyses. Along similar lines, Ln 246-247 – maybe raise this point initially, so the reader can follow the concordance.

Tables & Figures
- Tables 1 & 3 – consider indicating (e.g. via colour shading or grouping under sub-headings) which sites correspond to which genetic clusters (West, North, South).
- Fig. 2 – Is it possible to include % loadings on PCA axes to indicate the variation? Maybe also consider incorporating colours on the points, or ellipses; some way to indicate the clusters coincide with West, North and South nuDNA clades.
- Fig. 3 – Probably could be in supplementary material.
- Fig. 4 – Slightly confusing information display, with too much division of unclear units. Without more information regarding what the three clusters or panels correspond to location-wise the division seems either arbitrary or slightly misleading. If possible, perhaps have all panels connected.
- Fig. 5 – Fairly confusing and possibly not critical for the main text. The associated caption was also challenging; perhaps could be clearer if names in legend indicated the genetic data involved. Would be great if it was evident where these admixed individuals are with respect to the broader clade distributions on the map. Would it be possible to have the different maps – Figs. 1 & 5, even Fig. S1 – all incorporated into a single figure? Either a single map that indicates mtDNA and nuDNA clade limits, and perhaps admixed individuals, and the 16 sites (may be getting too busy…), or alternatively a figure with the different map displays featured as separate panels?
- Fig. 6 – could do with labeling the mtDNA clades West and East so as not to be misled by the variation in coloured symbols causing it to appear that there is no reason to the structuring. How many samples are represented in either clade (i.e. how many samples comprise the 10 West haplotypes)?

Minor comments:
• Add a comma: Ln 37 – after ‘urbanization’, Ln 40 – after ‘fragmented’, Ln 220 – “similar across sites, with”
• Spell out numbers: Ln 62 – ‘nine of ten’, Ln 215 – ‘six’, Ln 299 – ‘seven’
• Ln 70 – Add full-stop after ‘spp’
• Ln 91 – could samples be elaborated here to indicate what type of sample, e.g. tissue samples? It is clear a few sentences later that some are tissues, but for readers who are unsure what type of sample is being collected by the cloacal swab method, this would be useful.
• May not always be critical, but consider spelling out acronyms at first mention, e.g. Ln 94 – IACUC and TCU, Ln 96 – EPA
• Ln 117 – “and electrophoresed on an ABI”
• Ln 132 – “manufacturer’s protocols”
• Ln 136 – Add reference/citation for Sequencher
• Ln 185 – “MrBayes”
• Ln 224 – This seems like a very low Fst value regardless of p-value, if an Fst of zero implies complete panmixis.
• Ln 241 – Fig. S3 appears to be missing? Elephant Mountain WMA is mentioned in the text but not featured in any tables or figures.
• Add semicolon to maintain consistent style: Ln 242 – after ‘83%’, Ln 243 – after ‘95%’.
• Ln 259 – move ‘(q > 0.9)’ after ‘ancestry’ to minimize interruption to sentence flow. Could also remove ‘(q > 0.90)’ from Ln 261 as already defined at Ln 259. Furthermore, is ‘high ancestry’ the standard wording used with these types of measures; would it be clearer to say, ‘high ancestry assignment’?
• Could remove ‘horned lizards’ from Ln 263 and 264 if desired as not strictly necessary.
• Ln 275 – consider alternative word to ‘diverse’ such as ‘high’. Sounds odd as a describer of diversity.
• Ln 293 – maybe be cautious in referring to areas as hybrid zones or discussing hybridization, if within a species (at least as currently recognized) it is probably considered a region of admixture between populations.
• Ln 297-298 – Does Fig. 6 support this sentence? The wording is a little confusing.
• Ln 313-318 – from “There was a well-supported clade…” to “…also found in some of the same areas.” – These lines may not be necessary. The level of divergence between these groups of haplotypes is minimal and discussion of them potentially clouds the stronger findings that you want readers to take away.
• Ln 328 – some readers would expect a P-value to be provided with a statement indicating any “significant” statistics, e.g. “(P < 0.05)” or similar.
• Ln 337 – keeping consistent style, suggest move ‘(eastern and western)’ to end of sentence after ‘mtDNA loci’.
• Ln 343 – a little pedantic, but maybe rephrase this sentence – the presence of a lake poses a potential biogeographic barrier, but the formation of the lake would be the possible vicariant event.
• Ln 361 – suggest “that is also broadly concordant” or something similar. Consider the presence of northern microsat cluster individuals with eastern and western mtDNA haplotypes?
• Ln 364 – clarify what type of genetic distance, e.g. mtDNA? And what is this level high compared to; what are the divergence levels like between closely related species?
• Ln 410 – consistent unit notation, ‘~800 m’
• Ln 417 – higher or deeper?
• Ln 442 – “expected to come into”
• Ln 464 – Is “strong genetic breaks” reasonable wording, given the level of admixture occurrence? Seems like broadly concordant genetic disjunction, but strong breaks almost implies absolute deep reciprocal monophyly.
• Ln 481 – This sentence is great and an intriguing area to be considering further. If the authors are predicting that different habitats do indeed play a role in shaping divergence and structuring diversity (hinted at in the mention of xeric vs. mesic regions), it could be expected that differences in morphology (e.g. overall body size, scale size) and physiology (e.g. thermal tolerance, water balance) might also arise as mechanisms for dealing with pressures of water loss or extreme temperature variations.

---

## Round 0.2 · Minor Revisions

Thank you for your careful attention to reviewer comments and improvements to this manuscript.

In addition to comments raised by Reviewer 2, please address the following in your revisions:

1. Line 151 - please state whether the PCR products were sequenced in both directions or not.
2. Lines 426-432 and 469-471 It is unclear how the data presented are valid evidence of sex-biased dispersal. Please remove reference to sex-biased dispersal if explicit analyses are not conducted.
3. Table 2 - should "Warm Deserts" have shading in the "North" cluster?
4. Please check terminology for consistency (e.g. are the clusters "Northern, Western, Southern" or "North, West, South"?).

·

Basic reporting

The authors have done a great job incorporating further information into the Introduction to better set the scene as to the prior genetic knowledge available on the focal species, which emphasises the need and importance of their study. Likewise, the manuscript is much enhanced by the addition of detailed information on the environmental differences of the broad ecoregions inhabited by the different genetic divisions within the Texas horned lizard. Other minor changes throughout have also further improved the clarity and ease of following which genetic divisions are being discussed. The simple addition of genetic cluster identity within Tables 1 & 3 aid a lot in understanding and quick comparison of statistics across clusters. Adjustments to figures, particularly Figs. 1,3, & 4, have very much improved clarity; though there are still a couple of quite minor details that could be altered.

Experimental design

As mentioned above, additional background in the Introduction section now clearly identifies the knowledge gaps this study fills, adding to the importance of this work. Further minor adjustments and additions to the Methods section have also further improved clarity of exactly what was done during data collection and analysis.

Validity of the findings

The additional information and clarification added to the Discussion regarding differences in ecoregions, and possibilities for adaptations &/or fitness costs associated with different genetic groups and their crossings, adds to the conclusions and associated management recommendations made by the authors. It is clear from the added information that this work sets the stage for future avenues of study to investigate these issues.

Additional comments

Minor comments:

• Add a comma: Ln 33 – after ‘clusters’, Ln 125 – after ‘populations’, Ln 220 – after ‘stable’, Ln 283 – after ‘PCA’, Ln 294 – after ‘Plains’, and Ln 328 – after ‘haplotypes’.
• Change comma to semicolon: Ln 129 & 147 – after ’15 min’, Ln 130 & 148 – after ‘72°C’, and Ln 431 – after ‘Haldane’s rule’.
• Spell out numbers: Ln 190 – ‘ten independent’, Ln 193 – ‘ten runs’, Ln 321 & 344 – ‘ten unique’, and Table 1 & 2 captions – ‘ten microsatellite’.
• For consistency, either remove ‘v’ for version at Ln 135 (Genemapper), Ln 154 (Sequencher), Ln 215 (MrBayes), Ln 222 (DnaSP), or add to all other cases where the version number is provided for a program.
• For clarity, Ln 168 – add ‘(Ho and He)’ after ‘heterozygosity’, Ln 169 – add ‘(AR)’ after ‘richness’, and Ln 205 – add ‘(HR)’ after ‘richness’.
• Could remove ‘6.5’ after GenAlEx at Ln 177, 180, 203, & 207, as already listed at Ln 167; and ‘10’ after MEGA at Ln 210 & 211, as listed at Ln 154.
• Ln 29 & 529 – “…especially between the western and eastern clades…”
• Ln 143 – add a space between the hyphen and primer sequence for Leu.
• Ln 152 – Could remove “(ThermoFisher Scientific, Waltham, Massachusetts, USA)” as already listed at Ln 134.
• Ln 160, & Fig. 2 & 3 captions – for consistency in formatting, add a space between ≥ and 10.
• Ln 181 – “… STRUCTURE results (see below) which utilized all samples”.
• Ln 185 – for consistency, change ‘100,000’ to ‘104’, or change ‘106’ to ‘1,000,000’; also check Ln 217 depending on format decision.
• Ln 187 – ‘(K; Puechmaille, 2016; …’
• Ln 189 – Change ‘ALPHA’ to ‘α’ or ‘α value’, since α was previously used in this sentence.
• Ln 208 – as was explained to reviewers, perhaps add something to the end of this sentence as justification for not concatenating mtDNA loci, e.g. “… and the control region due to the difference in sampling for each marker”.
• Ln 212 – ‘branch swap filter’.
• Ln 222 & 224 – ‘DnaSP’.
• Ln 223, 311, 336 (x3) – italicize the h for haplotype diversity ‘(h)’.
• Ln 230 – Could just have either ‘tau’ or ‘τ’ given it is listed in Ln 228.
• Ln 287 – Could remove ‘ecoregion’ after ‘Southern Texas Plains’?
• Ln 311-314 – It took a few reads and looking back at Tables 1 & 3 to understand what these sentences were getting at. I would suggest at least referring the reader to both tables (Table 1, 3) either at the end of the sentence at Ln 312 or Ln 314 so it is clear that values from both should be compared to get this point.
• Ln 322-324, 326, 345-348, & Table 1 NA column – consistency of number of decimal places quoted. If standard error is to two decimal places, mean should also be to two decimal places, similarly with ranges.
• Around Ln 324 – Could you maybe include the divergence between P. cornutum and P. blainvillii, like is provided in the ND4 section for P. asio?
• Ln 343, & Fig. 5 & 6 captions – remove hyphen in ‘maximum likelihood’.
• Ln 346 – Remove ‘=’ and the space between ‘1.7’ and ‘%’.
• Ln 359 – would again provide a P-value with the significantly negative statement, simply (P < 0.05), and reference (Table 4) at the end of this sentence regarding Tajima’s D results.
• Ln 368 – Minor but consider reordering the listing about mtDNA groupings (western and eastern) for consistency, this reinforces in the reader’s mind that west matches west and east matches north + south. Often throughout the text west is discussed before east because it is the more straightforward division.
• Ln 371 paragraph – This section is excellent. You could even remind readers as it goes which genetic cluster is associated with each region being discussed, for example, Ln 377: “The Chihuahua desert ecoregion (west clade)”, Ln 380: “The South-Central Semi-Arid Prairies ecoregion (northern cluster)”, etc. But that is not at all critical and it would be perfectly fine without also.
• Ln 454 – (MUs; Moritz, 1994)
• Ln 463-4 – If citing chronologically, swap Endriss, 2006 and Stark et al. 2005.
• Ln 476 – ‘effective population size’?
• Table 2 caption – remove comma after (q).
• Fig. 1 & 2 – These figures are much improved and easy to understand. It would be nice if it were possible to have them combined in a single page figure somehow as two panels, Fig. 2 doesn’t even necessarily have to be so large. But it would be understandable if this forced a reduction in size of either map that would compromise the clarity of the information.
• Fig. 3 caption – state somewhere in this caption that these values are from the nuclear microsats to ensure it’s very clear to readers what data the values are from. It also says, “dark grey and red on #1 indicates…”, should there be grey on the symbol for site 1 which is not currently there?
• Fig. 4 caption – again, state something like “… clustering of nuclear multilocus genotypes…”
• Fig. 5 caption – ‘well-supported’. Are the nodes equal to 80 or equal to/greater than? Only one clade seems to be indicated as being supported with 80 bootstraps. It is tricky to see because of the circular format of this tree where exactly the eastern clade is delineated. It would likely be much easier to distinguish the clades and branchlengths if the tree were presented like that in Fig. 6. A bootstrap value of 80 however could be argued to not be a particular well-supported node, and if there is not a clearly supported node at the base of the other clade then the results section for this particular locus should probably be updated to be clear that there appear to be two clades, east and west, but that neither is strongly supported by this particular analysis.
• Fig. 5 & 6 – whilst it now feels clearer that the symbols indicate the different nuclear clusters both figures would be further enhanced by clearly labelling the two mtDNA clades somehow. For example, in Fig. 6, the branches leading to either clade could be labelled as ‘West’ and ‘East’, or alternatively could have labelled vertical bars to the right of each clade.

---

## Round 0.3 · accepted · Accept

Thank you for your attention to detail in your revisions!